# Can sterilization of disposable face masks be an alternative for imported face masks? A nationwide field study including 19 sterilization departments and 471 imported brand types during COVID-19 shortages

**B. van Straten**[1]*, **P. D. Robertson**[1], **H. Oussoren**[2], **S. Pereira Espindola**[3], **E. Ghanbari**[3], **J. Dankelman**[1], **S. Picken**[3], **T. Horeman**[1]

**1** Department of BioMechanical Engineering, Delft University of Technology, Delft, The Netherlands, **2** Amsterdam University Medical Center, Amsterdam, The Netherlands, **3** Department of Chemical Engineering, Delft University of Technology, Delft, The Netherlands

\* b.j.vanstraten@tudelft.nl

## Abstract

### Background

Face masks, also referred to as half masks, are essential to protect healthcare professionals working in close contact with patients with COVID-19-related symptoms. Because of the Corona material shortages, healthcare institutions sought an approach to reuse face masks or to purchase new, imported masks. The filter quality of these masks remained unclear. Therefore, the aim of this study was to assess the quality of sterilized and imported FFP2/KN95 face masks.

### Methods

A 48-minute steam sterilization process of single-use FFP2/KN95 face masks with a 15 minute holding time at 121˚C was developed, validated and implemented in the Central Sterilization Departments (CSSD) of 19 different hospitals. Masks sterilized by steam and $H_2O_2$ plasma as well as new, imported masks were tested for particle filtration efficiency (PFE) and pressure drop in a custom-made test setup.

### Results

The results of 84 masks tested on the PFE dry particle test setup showed differences of 2.3 ±2% (mean±SD). Test data showed that the mean PFE values of 444 sterilized FFP2 face masks from the 19 CSSDs were 90±11% (mean±SD), and those of 474 new, imported KN95/FFP2 face masks were 83±16% (mean±SD). Differences in PFE of masks received from different sterilization departments were found.

**Data Availability Statement:** All relevant data are within the manuscript and its Supporting information files.

**Funding:** The author(s) received no specific funding for this work.

**Competing interests:** The authors have declared that no competing interests exist.

## Conclusion

Face masks can be reprocessed with 121 ˚C steam or $H_2O_2$ plasma sterilization with a minimal reduction in PFE. PFE comparison between filter material of sterilized masks and new, imported masks indicates that the filter material of most reprocessed masks of high quality brands can outperform new, imported face masks of unknown brands. Although the PFE of tested face masks from different sterilization departments remained efficient, using different types of sterilization equipment, can result in different PFE outcomes.

## Introduction

After the outbreak of COVID-19, this respiratory disease has spread at a rapid pace [1, 2]. Adequate face masks are essential to protect healthcare professionals. In many hospitals shortages of personal protection equipment occurred due to increased demand [3]. In the search for alternative sources, hospitals started to consider reusing their single-use face masks by sterilizing them [4].

Face masks, also referred to as half masks, are used to protect individuals against airborne particles during aerosol generating procedures. Three classes of particle filtering face piece (FFPs) are described in European Norm (EN) 149:2001+ A1:2009 [5]. The most commonly used masks in relation to COVID-19 are the Class 2 FFP2 masks. These are considered to be equivalent to the American N95 mask [6], conforming to the standards of the National Institute for Occupational Safety and Health (NIOSH) 42 CFR 84 mask [7], and the Chinese KN95 mask complying to the Guobiao (GB) 2626–2006 standard [8]. The filter efficiency of smaller particles is a crucial element. The European Norm requires a minimum filter efficiency of 94%, whereas NIOSH [7] and GB [8] require 95%.

### Testing filter material of face mask

EN 149:2001+A1:2009 [5] and more specifically NEN-EN 13274–7:2019, part 7 describe a test setup to determine the particle filtration efficiency (PFE) of face masks. It consists of a flow tube, a flow generator, a NaCl particle generator and two particle measurement devices and generates flows up to 120 l/min with NaCl particles of 0.1 to 10 μm. Unfortunately, this setup is costly to build. Therefore, in the first 2 months of COVID-19 only 2 systems were operational in the Netherlands that could be used for testing new, imported face masks. The cost for testing one face mask was approximately 1,500 Euro with a waiting list of up to 4 weeks. A new quick testing method was needed.

### Potential reprocessing methods

Several studies have shown the effects of different sterilization methods, including gamma sterilization, plasma sterilization, steam and dry heat sterilization, microwaves, washing machines and UV–C light, as methods to decontaminate face masks for reuse [9–13]. These studies suggest that gamma and steam sterilization conducted at 134˚C damage the microstructure of the filter material [9].

Washing machines and microwaves have a low capacity, and microwaves do not create a uniform heat distribution and require a steam bag [10–12]. Some studies suggest that the high concentration of liquid $H_2O_2$ in plasma sterilization (approx. 60%), and its strongly charged ionized vapor may neutralize the electrostatic charge of the filter media [11, 12]. Moreover, the

sterilization efficacy would likely be affected by the presence of moisture (e.g. exhaled breath) in worn masks, as water is a polar molecule. Finally, the capacity per run remains low due to the vacuum-driven process [13]. The evaporation of moisture may restrict the sterilizer's ability to pull deep vacuum. UV treatment of face masks seems to have potential but requires preparation time as face masks need to be unfolded in such a way that UV light reaches all of the mask material [10–13]. UV-B sterilization was not considered as this method is not yet commonly used and not readily available at hospital sterilization departments. Steam sterilization at 121˚C could be an option since studies have shown the effectiveness at 121˚C in inactivating the Coronavirus [14, 15].

Pilot studies, that included ATCC 12228 bacterial testing, have been conducted to determine whether 121˚C sterilization was a safe and effective method to deactivate the Corona virus. The protocols and results were made available to hospitals via the repository of the Delft University of Technology after demonstrating that sterilization of face masks was possible up to 5 times for high-quality face masks [9, 16]. Although proven efficient, the potential of this new 121˚C sterilization method was not explored. Moreover, a study where many different brands of face masks were processed at different CSSD's, with comparisons between new, imported masks and sterilized masks, did not exist. Therefore the aim of this study is to find the best alternative for high quality face masks in times of shortage by assessing the quality of sterilized and imported FFP2/KN95 face mask filter materials.

The following research questions were defined:

1. Can FFP2 masks be reprocessed using 121˚C steam or $H_2O_2$ plasma sterilization?

2. Are reprocessed face masks an alternative for new ones?

3. What effect does sterilization have on the materials?

## Methods

A sterilization facility of a Dutch CSSD (ISO 7 validated, Van Straten Medical, De Meern, the Netherlands, operated by CSA services) was set-up for the purpose of reprocessing used (potentially COVID-19 contaminated) FFP2 face masks. New testing methods were developed to test the filter material quality after sterilization [4, 9, 16]. The testing was carried out for any hospital, reseller or manufacturer wanting to check the quality of sterilized or new, imported face masks.

### Reprocessing by 121˚C steam at CSA services sterilization

Within this new reprocessing approach, decontamination was done solely by steam sterilization. To implement the 121˚C sterilization process, a special logistical routing was set-up to collect and process face masks. Upon receipt, the masks were removed from their double wrapping and inspected individually for visual damage. In case of deformations, dirt, lipstick, hairs, black streaks, stains or other deviations, the masks were discarded. The visually approved face masks were marked with a dot and packaged in autoclavable impermeable sterilization laminate bags (type CLFP150X300WI-S20, Halyard, UK) (Fig 1). A mask was disposed after it was marked with a maximum of 5 dots. A maximum of five face masks were packaged per bag to ensure proper sterilization. The autoclaves (GSS6713H-E, Getinge, Sweden) were activated with a 121˚C program and re-validated. The autoclave cycle was set for 48 minutes with a 15 min holding time (high vacuum 121 ˚C; ≥15 min HT, total CT 48 min). Face masks with a higher class (FFP3/N95) were treated as FFP2/KN95. The PFE for the average FFP2/

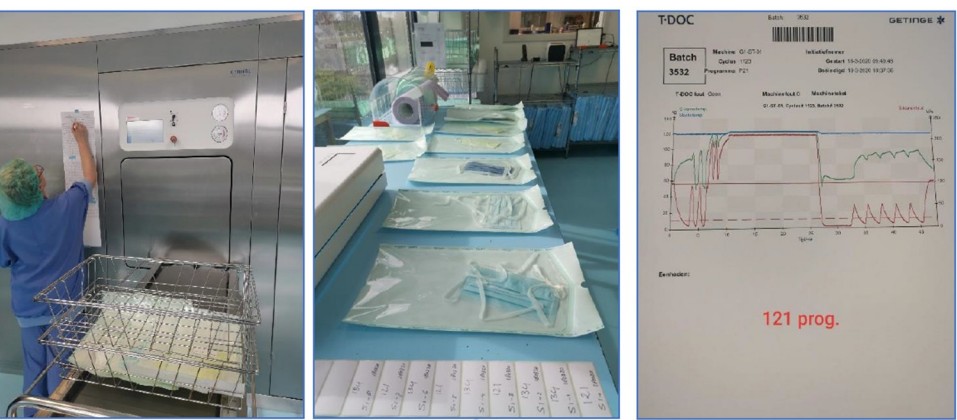

**Fig 1. Autoclave procedure with Halyard laminate bags.** Left, laminated bags entering the autoclave. Middle, masks are wrapped in laminate. Right, the 121 ˚C steam sterilization program as used for face mask sterilization.

KN95 mask material has to be 94% or higher for a pass and under 94% for a fail [5]. The performance of the mask material was determined by measuring the PFE and breathing resistance. Fig 2 shows the particle counter with a custom-made particle chamber (Lighthouse Solair 3100, San Francisco, www.golighthouse.com). The machine drew air through the mask into the chamber and to the particle counter. The diameter of the chamber was chosen such that it guaranteed sufficient airflow through the filter material for the particle counter [9, 16, 17]. The PFE was determined by measuring the difference in the number of particles before

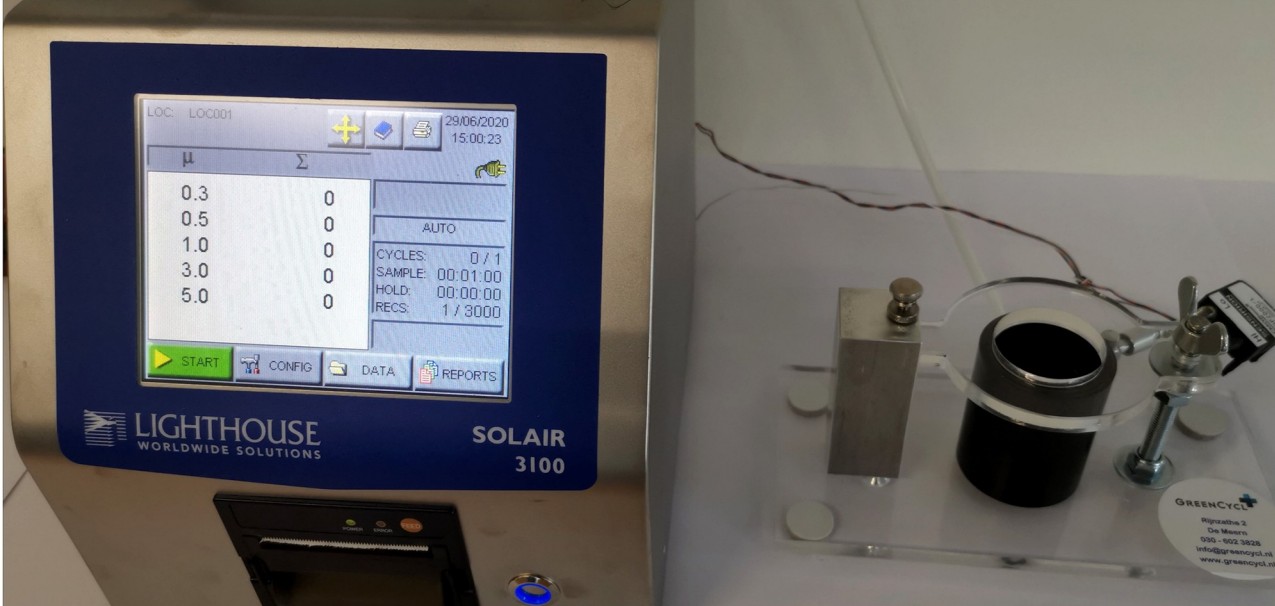

**Fig 2. Lighthouse Solair 3100 particle counter connected to a particle chamber.** When very dense filter materials are used with a very high PFE for the smallest particles, it causes a breathing resistance for the user [16, 17]. This resistance causes a pressure drop which was measured using an analog differential pressure sensor, type SDP2000-L (Sensirion AG, Staefa ZH, Switzerland) connected to the particle chamber. The pressure sensor is temperature compensated, calibrated and has a resolution of 11 Pa with a repeatability of 0.3% and accuracy of 1% [17]. The breathability requirements for respiratory protective devices are provided in a European standard [18]. The maximum permitted resistance (mbar) differs for the FFP1, FFP2, and FFP3 masks, ranging from 0.6–1.0 for inhalation at 30 l/min, 2.1–3.0 for 95 l/min and 3.0 for exhalation at 160 l/m. The norm for a FFP2-mask at 30 l/min is 0.7 mbar.

and after filtration by the mask. First, the particle concentration in a standard volume of room air was determined by measuring the number of particles (sizes 0.3, 0.5, 1.0, and 5.0 μm) in a volume of surrounding air. Second, the mask was installed on the chamber to measure the number of particles after filtration.

### Test setup validation to the European Norm EN 13274–7

The accuracy of the developed particle test setup was evaluated by comparing results from known face masks, tested on (our) particle setup, with the results of the same brand and type masks, tested on a continuous flow system. The continuous flow test system used NaCl particles and was built at the Delft University of Technology according to NEN-EN 13274–7:2019 [19].

Additionally, the standard EN 149:2001+A1:2009 describes experiments to determine if a mask creates a proper fit on the face without leakages. The inward leakage is determined by means of a fit test and strap test [17, 18]. In this study, inspection of the materials were conducted and leakage tests were performed on all reprocessed masks. Although this study focused on the material properties, only masks that showed no change in fit or material properties were included. The types that did deteriorate were registered and disposed after arrival. Although we followed the EN-149 standard as much as possible, we did reference our outcomes with the NaCl test since we used a custom-made test setup as a non-standard EN-149 methodology.

### 121 °C steam sterilization consistency between CSSDs

The consistency of sterilization results, caused by different processes and equipment was compared between 19 CSSDs. Samples of masks representing the most commonly used brands and types were selected and measured with the PFE setup. Only CSSDs were included that provided a minimum of four masks that were sterilized once. Face masks were not cleaned after visual inspection and prior to sterilization. A student's t-test (two tailed, unequal variance, SPSS 17.0) was used for comparison, and a probability of $p < 0.05$ was considered to be statistically significant.

### Face mask material differences

Differences in mask material were analysed by chemically and thermally comparing the fabric of the two most common types. Therefore, a differential scanning calorimetry (DSC), an X-ray diffraction (XRD) and a transform infrared spectroscopy (FTIR) were conducted (S1 File).

### Testing new masks

To determine how many samples of imported face masks were needed, the variance of the PFE and pressure drop was determined on three imported face masks. Ten measurements conducted on each face mask type (S2 File) showed that the largest variance was found in the 0.3 μm particle size category of 0.6%, 1.1% and 0.3% of the mean values respectively. The pressure drop measurements showed a variance of 0.7%, 1.8% and 1.8% of the mean values respectively. This low variance indicated consistent behavior of the filter materials. Combined with the importance of a short processing time it was determined to measure a minimum of two masks of each type that was provided by the clients. The averaged values are listed in S3 File. In case a deviation of more than 10% was found in the 0.3 μm category, two additional masks were tested and the supplier was notified. This data was excluded from the study.

Samples were selected for PFE measurement from batches of imported masks. The PFE results of those new face masks were compared to the PFE results of the sterilized face masks from the 19 CSSDs. New imported face masks that scored above 98% PFE in the particle range were further investigated for breathability by measuring the pressure drop.

## Results

### Reprocessing by 121˚C steam sterilization at CSA services

A total of 74,834 masks were processed by the CSSD of CSA services. Of these masks, 56,668 were disposed after incoming inspection due to visual damage, deformities or dirt. The remaining 18,166 face masks were steam sterilized at 121 ˚C. Table 1 shows the top five brands that were sterilized and returned to hospitals for use.

**Test setup validation to the European Norm EN 13274–7.** Preliminary tests conducted with 84 different masks, tested on the PFE dry particle test setup and a NaCl test setup, following the EN 13274–7:2019 standard, indicated an outcome deviation of 2.3±2% (mean±SD) on average with a max of 7% (S4 File). A measurement test conducted with another ten different masks indicated that an average of 19 s ±21% (mean±SD) (SD 21) is needed to install and inspect the mask on the particle counter and an additional 15 s ±13% (mean±SD) is needed to take the mask from the system after 1 minute of measurement. None of the masks showed visual signs of deformation or damage after being measured.

**121 ˚C steam sterilization consistency between CSSDs.** The reprocessing methods by means of steam sterilization were adopted by 19 hospitals (Amsterdam University Medical Center (VUmc and AMC locations), Holendrecht Medical Center, Franciscus Hospital, CombiSter RDGG & Haga, Spaarne Hospital, Erasmuc MC, University Medical Center Groningen, Leiden University Medical Center, Flevo Hospital, Isala Hospital, Diakonessenhuis Utrecht, VieCuri, Rode Kruis Hospital, Noordwest Hospital Group, Amphia Hospital, and Tweesteden Hospital). The PFE results of 444 reprocessed FFP2/KN95 face masks from the CSSDs of 19 different hospitals in the Netherlands are provided in Fig 3. Of the 444 masks, 371 masks were sterilized with steam sterilization and 73 with $H_2O_2$ plasma (S5 File).

From these 444 tested face masks, 58 3M 1862+ face masks were provided by seven CSSDs of four university hospitals, one one general hospital and one general practitioner which were only sterilized once at 121 ˚C using steam sterilization (S6 File). The influence of different installations, protocols or staff on the PFE is shown in Fig 4. The "N" value indicates how many 3M 1862+ face masks were included in the study that were only sterilized once. The statistical tests reveal differences in outcome mainly for the CSSD of University Hospital 2.

**Face mask material differences.** The 444 face masks consisted of 101 different types. Of the 101 different types, 3M 1862 and Kolmi Op-Air were mostly tested. The PFE results of 89 3M 1862 and 26 Kolmi Op-Air are provided in Table 2 for 0.3, 0.5, 1 and 5 μm particles.

**Table 1. Top five reprocessed face masks.**

| Brand (Type) | Percentage |
|---|---|
| 3M (1862+) | 42% |
| 3M (1872+) | 21% |
| My-T-Gear | 8% |
| IMG Europe (R620) | 5% |
| Kimberly Clarc Corp | 5% |
| Rest | 19% |

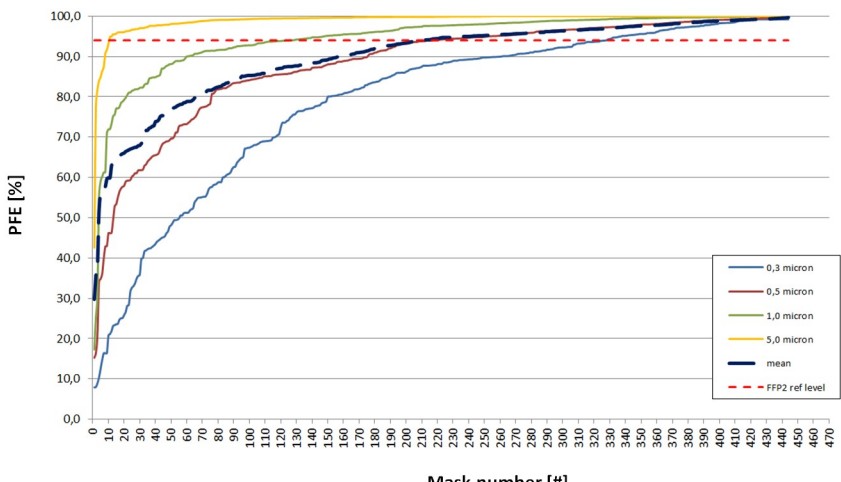

**Fig 3. PFE values after sterilization with 121 ˚C steam or $H_2O_2$ plasma sterilization in chronological order from worst to best.** The red dotted line indicates the FFP2 level at 94% PFE. Each mask number represents a sample of a sterilized batch from one type only.

The results indicate that 3M 1862 shows low PFE values after 2x $H_2O_2$ plasma processing and that Kolmi Op-Air shows low and inconsistent PFE values after 1x 121 ˚C processing (S3 File).

**Thermal properties of 3M Aura 1862+ and Kolmi Op-Air M52010 face masks using DSC.** The three tests, differential scanning calorimetry (DSC), X-ray diffraction (XRD) and transform infrared spectroscopy (FTIR), confirmed that both masks consisted of 5 layers with the profile of Polypropylene (PP) material (S7 File).

**Tests of new, imported masks.** The PFE results of 471 different types of new, imported FFP2/KN95 face masks from collaborating hospitals and resellers are shown in Fig 5. Of these, 27 face masks scored above 98% PFE for the 0.3 micron particle size category. These masks were tested for breathability by measuring the pressure drop (S8 File). Fig 6 shows the breathing potential of the 27 face masks. The material of 27 face masks with high PFE values showed pressure drops between 251 and 3976 Pa on the measurement setup. When calculated for the total mask areas A and B, five out of 27 masks showed a total pressure drop higher than the EU

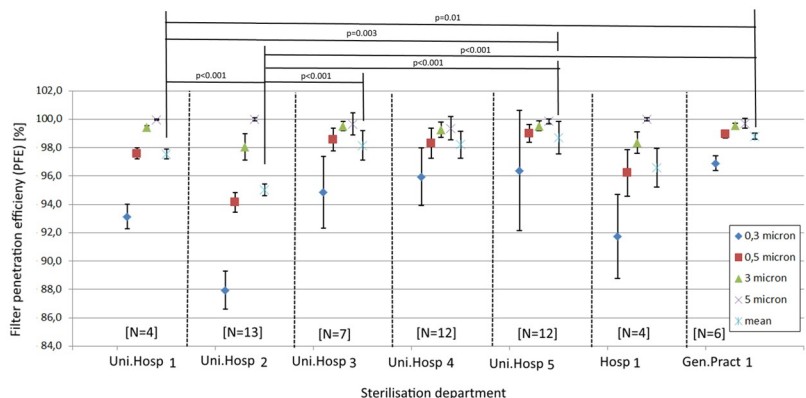

**Fig 4. PFE values with standard deviation of different 3M 1862+ coming from 7 different CSSDs.** Statistical differences are indicated with P values above the figure.

**Table 2. Particle filter efficiency of two commonly used mask after either 121 ˚C steam or H₂O₂ plasma sterilization.**

| Brand & type | Number of masks | Sterilization method | 0.3 μ | 0.5 μ | 1 μ | 5 μ | Mean |
|---|---|---|---|---|---|---|---|
| | | | % PFE (SD) | % PFE (SD) | % PFE (SD) | % PFE (SD) | % PFE |
| **3M 1862** | 5 | H₂O₂ | 86,4 | 93,8 | 97,4 | 99,5 | **94** |
| | | Sterrad | (12,5) | (6,2) | (2,7) | (0,5) | |
| **3M 1862** | 72 | 121 ˚C | 93,6 | 97,3 | 99,0 | 99,7 | **97** |
| | | steam | (4,1) | (2,1) | (0,8) | (0,7) | |
| **3M 1862** | 4 | 2 x H₂O₂ | 41,3 | 66,9 | 83,9 | 99,5 | **73** |
| | | Sterrad | (1,7) | (1,6) | (1,3) | (0,4) | |
| **3M 1862** | 8 | 2 x 121 ˚C | 91,6 | 96,2 | 98,3 | 100 | **97** |
| | | steam | (3,2) | (1,8) | (0,8) | (0,1) | |
| **Kolmi OP-Air M52010** | 11 | H₂O₂ | 89,8 | 96,4 | 98,4 | 99,8 | **96** |
| | | Sterrad | (1,4) | (1,4) | (0,5) | (0,3) | |
| **Kolmi OP-Air M52010** | 15 | 121 ˚C | 21,2 | 56,3 | 78,4 | 99,8 | **64** |
| | | steam | (6,8) | (8,5) | (8,2) | (0,5) | |

standard of 0.7 mbar [18]. Finally, four masks showed readings at approximately 3.7 mbar, which is very close to the maximum measurable pressure drop of 4500 Pa. In two occasions the PFE data of the two tested masks of the same type deviated more than 10% (i.e. PFE of 67% vs 84% at 0.3 μm). Closer inspection revealed that the two masks had a different appearance as one had an additional logo in the shape of a heart.

## Discussion

Regarding the research questions, it can be confirmed that FFP2 masks can be safely reprocessed with 121 ˚C steam sterilization if appropriate testing facilities are available. The data from Figs 3 and 5 indicate that reprocessed face masks can act as alternatives for new face masks as sterilization of well-known brand often gives better PFE results compared to newly imported masks. Although the base materials are similar, the manufacturing, preparation, and use of coatings have a large effect on the PFE of mainly the smaller particles.

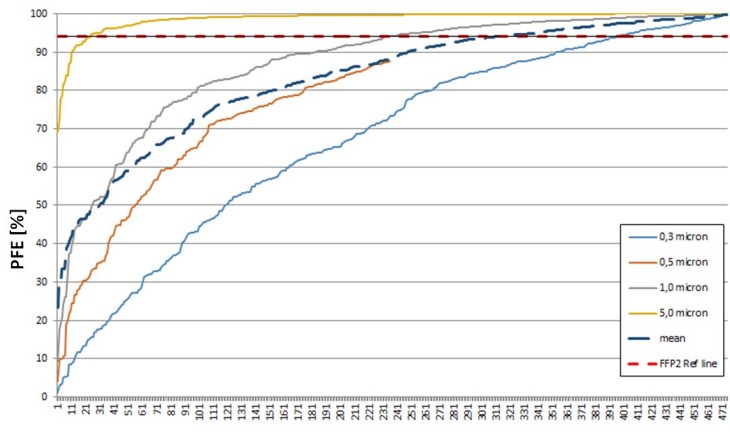

**Fig 5. PFE values of new imported FFP2/KN95 face masks in order from worst to best.** The red dotted line indicates the FFP2 level at 94% PFE. Each mask number represents a sample of a new batch from one type only.

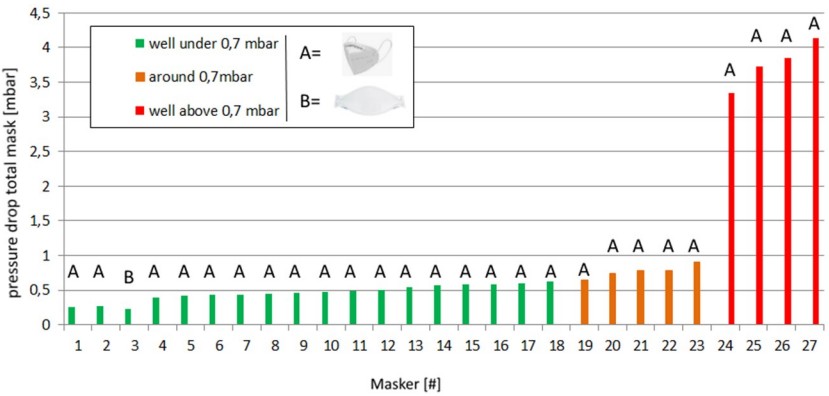

**Fig 6. Pressure drop of 27 new face mass with PFE>0,7 mbar.** Four masks performed really low (red), 5 performed around the EU norm of 0,7 mbar and 18 performed well according to the EU norm of 0,7 mbar.

The cross validation with the NaCl continuous flow setup built according to EN 13274–7 standard showed that the most important requirements for determining the filter material properties were met. After nineteen hospitals adopted the steam sterilization process, a nation-wide data field experiment was initiated that informed multiple international NGOs, universities, and industry members about the advantages and disadvantages of sterilization of face masks [17–21], which led to a Dutch standard for sterilization of face masks. After the first results were shared by request [22], general practitioners, dental practices and pharmacies claimed to successfully adopt the 121 ˚C sterilization process in their smaller sized autoclaves with sufficient results [16].

Sterilization with the purpose of reusing medical devices is often driven by cost savings [23]. However, some studies also report the reuse of medical devices to realize environmental benefits [24]. In this study, steam sterilization is used to prevent shortages. In 1986, a survey was conducted including Canadian hospitals reusing disposable medical devices [25]. Forty-one percent of the hospitals confirmed that they reused disposable medical devices, with respiratory therapy equipment as the most reused medical device.

Testing by particle counting seems to be essential for both new and sterilized single-use face masks as it indicates the quality of the mask in terms of filtration capacity. This became evident as our data revealed large differences in PFE despite the similar appearance of the mask material. Our results in S6 and S8 Files indicate the presence of coatings that improve the electrostatic behaviour of the mask. As the presence of these kinds of coatings is very difficult to demonstrate, it is advised to test the PFE with a particle counter at all times. To rule out that reprocessed and new face masks do not meet the stated FFP standard, a particle test as a 'quick and dirty' test could be applied on every batch, as the test method described in this study can give a quick indication of the quality.

As high quality FFP2 masks react differently to different sterilization methods, it is expected that the electrostatic charge of a mask has a major effect on the PFE especially for smaller lighter particles. Although not part of this study, it would be interesting to investigate how either 121 ˚C sterilization or $H_2O_2$ plasma affects the mask's electrostatic charges and how this is related to the fiber orientation, pore size and openings between the stacked layers.

Face masks sterilized with the intention of reuse could furthermore undergo a "fit test". This test may be regarded as a fit validation conforming to a proper fit on the face without leakages around the mask. To assure a decreased risk of spreading other diseases, the bio efficacy of a face mask should also be considered. Tests regarding this aspect were conducted

previously and appeared negative for bacteria on steam-sterilized face masks that were tested at the dept. of Microbiology at Franciscus Hospital in the Netherlands [9].

Testing face masks for particles is important in the quality assurance of the sterilization process. Our data shows that despite the implementation of similar 121 ˚C sterilization protocols, mean PFE outcomes can differ up to 6%. As the types of masks and sterilization methods are similar, the only unknown variable is the wearing/processing influence on the mask during use, transport and inspection. University Hospital 2 in Fig 4 seems to show much lower PFE outcomes. It could be that stretching and bending of the mask can influence the integrity. However, it is also expected that the confidence interval would have been larger as the intensity of the stretching and bending is human dependent. As the confidence interval of the mean PFE outcomes of University Hospital 2 seems similar or even smaller than those of other hospitals, it is advisable to perform validation tests at all hospitals.

In the CSSD at De Meern, a 10% tolerance was accepted for sterilized face masks after testing. Therefore, an 84% filtration capacity on a 0.3 μm particle level was the minimum limit. Although not based on any evidence in the literature, this percentage was considered to be sufficient with respect to the shortages of face masks, taking into consideration that the Coronavirus (SARS-CoV-2) is mainly spread through 0.3 μm or larger droplets. However, a consensus needs to be made to actually define the minimal allowable PFE values in times of crisis.

The DSC, XRD and FTIR test results in S6 File conducted on each of the five layers of the 3M Aura 1862+ and Kolmi op-Air M52010 masks reveal that all layers are made of the same Polypropylene material. The differences in behavior when sterilized cannot be explained by chemical composition. A detailed interpretation of the results can be found in S9 File.

The data of 410 sterilized and 471 newly imported KN/N95 or FFP2 face masks reveal that, despite the differences in PFE between different sterilization processes, still approximately 75% of the face masks of known brands reach the FFP2 standard after sterilization when compared to only 50% of newly imported, less known brands. Our results suggest that the technology needed to manufacture a good mask is not easy. Manufacturing and quality assurance should be monitored and controlled by the government. During the study period, it was observed two times that within a single batch of imported face masks, the quality and layout of the masks were different, despite being wrapped in the same packaging with the same printed PFE standard. This suggests that multiple factories were supplying to one brand. In other cases, some masks (Fig 6) showed almost complete lack of air penetration due to the use of wrong materials or manufacturing processes.

Although our results indicate that sterilized face masks can be used if the filter material can be properly tested, it might be considered that wearing a used mask can have a psychological impact on healthcare workers. To overcome this issue, masks could be marked with the user's initials so that it can be returned to the same person.

## Study limitations

It is of utmost importance that the reprocessing of single-use PPE, as described in this study, is equivalent to existing standards. Each deviation or omission of such standards needs a clear demonstrated equivalence with the applying standards. In our setup, only environmentally dry particles were used in the developed rapid test setup. Although we validated the dry particle setup with an aerosol testing setup (NaCl test, paraffin oil setup) according to the EN 13274–7 standard, it was only possible to compare the PFE for a limited range of particle sizes. Therefore, in-depth knowledge about the PFE related to particle size was not generated. To identify other potential differences between the dry particle and continuous flow setups, a 'gap' analysis should be conducted. Other than testing the basic material of the filter layers, we were not able

to indicate the presence of surface active coatings. Therefore, it was not possible to investigate the role of surface active coatings on the melting or oxidation of the fibres. Although the study used a validated reprocessing method based on 121˚C sterilization to inactivate the virus in the mask, the retention of the inactivated virus has not been studied and should be investigated further in future studies.

## Conclusions

Sterilization of disposable face masks by means of standardized steam sterilization at 121˚C could be an alternative during face mask shortages due to COVID-19 as long as the fit does not change and the filter materials are not significantly affected by heat. The varying efficiency after reprocessing amongst different brands shows that only quality masks of particular brands such as 3M Aura 1862, 3M Aura 1873 and My-T-Gear 301 are suitable for limited reuse. The data show that the 121˚C sterilization process can be safely implemented as long as proper testing of each batch is possible and the process and logistics is well controlled. The new PFE testing method proved to be accurate enough to determine degeneration of the mask material after sterilization and to determine the material quality of imported face masks. FTIR, XRD and DSC measurements indicate that all layers from both masks are made from Polypropylene. Future testing is needed to determine if differences in PFE outcomes after sterilization with 121 ˚C steam or $H_2O_2$ plasma can be explained by the level of crystallinity, or by the orientation and dimensions of the fibres and potential proprietary treatment in the layers of the face mask. PFE comparison between sterilized masks and imported face masks with varying filter qualities indicates that health care professionals in some cases can better reuse a known reprocessed brand rather than an imported face mask from a reseller with an unknown brand.

## Supporting information

**S1 File. DSC, XRD and FTIR tests on face mask materials.**
(PDF)

**S2 File. Measurement reports.**
(PDF)

**S3 File. Validation PFE outcome of particle counter setup against continues flow system of Delft.**
(PDF)

**S4 File. Particle counter, MISIT.**
(PDF)

**S5 File. Comparison between CSSD's.**
(PDF)

**S6 File. New foreign masks tested on particle counter setup.**
(PDF)

**S7 File. Material analysis.**
(PDF)

**S8 File. Testing pressure drop over filter material.**
(PDF)

**S9 File. Discussion material test results.**
(PDF)

## Acknowledgments

The authors would like to thank the CSSD personnel and sterilization specialists from all participating hospitals for sending us samples of their sterilized batches for testing. Special thanks to the personnel and management from CSA Services for transforming their CSSD into a high volume face mask processing line. The students and colleagues of TU-Delft who helped test during 'ProjectMask' are thanked for their effort. Finally, resellers are thanked for testing their masks at our test facility at De Meern and Delft University of Technology.

## Author Contributions

**Conceptualization:** B. van Straten, J. Dankelman, S. Picken, T. Horeman.

**Data curation:** B. van Straten, P. D. Robertson, H. Oussoren, E. Ghanbari, S. Picken, T. Horeman.

**Formal analysis:** B. van Straten, P. D. Robertson, S. Pereira Espindola, E. Ghanbari, J. Dankelman, S. Picken, T. Horeman.

**Investigation:** B. van Straten, P. D. Robertson, S. Pereira Espindola, E. Ghanbari, J. Dankelman, T. Horeman.

**Methodology:** B. van Straten, S. Pereira Espindola, E. Ghanbari, J. Dankelman, S. Picken, T. Horeman.

**Project administration:** B. van Straten, P. D. Robertson, H. Oussoren, S. Pereira Espindola, E. Ghanbari, T. Horeman.

**Supervision:** J. Dankelman, T. Horeman.

**Validation:** T. Horeman.

**Visualization:** T. Horeman.

**Writing – original draft:** B. van Straten, J. Dankelman, T. Horeman.

**Writing – review & editing:** B. van Straten, J. Dankelman, T. Horeman.

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
