## [Decision Letter · Decision Letter 0]

27 Jul 2021

PONE-D-21-13083

Sterilization of disposable face masks with respect to COVID-19 shortages: a nationwide field study including 19 sterilisation departments

PLOS ONE

Dear Dr. van Straten,

Thank you for submitting your manuscript to PLOS ONE. After careful consideration, we feel that it has merit but does not fully meet PLOS ONE’s publication criteria as it currently stands. Therefore, we invite you to submit a revised version of the manuscript that addresses the points raised during the review process. Reviewer's comments are appended below.

We look forward to receiving your revised manuscript.

Kind regards,

Mahendra Singh Dhaka, Ph.D.

Academic Editor

PLOS ONE

Journal Requirements:

2. In your Methods section, please clarify how imported masks were selected to be included in the analysis; and how many samples were tested for each brand

Reviewers' comments:

Reviewer's Responses to Questions

**Comments to the Author**

1. Is the manuscript technically sound, and do the data support the conclusions?

Reviewer #1: Yes

Reviewer #2: No

Reviewer #3: Partly

2. Has the statistical analysis been performed appropriately and rigorously? 

Reviewer #1: Yes

Reviewer #2: No

Reviewer #3: Yes

3. Have the authors made all data underlying the findings in their manuscript fully available?

Reviewer #1: Yes

Reviewer #2: Yes

Reviewer #3: Yes

4. Is the manuscript presented in an intelligible fashion and written in standard English?

Reviewer #1: Yes

Reviewer #2: Yes

Reviewer #3: No

5. Review Comments to the Author

Reviewer #1: The present manuscript discusses about the efficiency of different methods to reuse the masks, which is the critical need of the hour and can save both, the time and money during this pandemic time. The manuscript, in my opinion is acceptable for publication.

Reviewer #2: The test conducted in this paper tests the used face masks for particle size range 0.3 to 5 micron. The low efficiency of the processed face masks at 0.3 microns is concerning and contradicts the conclusion. Furthermore, the difference in efficiency after reprocessing among the different brands concludes that only masks of a particular brand could be reused.

I am also concerned about the sample size and whether it accurately represents the entire population. There is no sample size test conducted to identify the required sample size to detect the variance to be checked.

Reviewer #3: The manuscript titled “sterilization of disposable face masks with respect to COVID-19 shortage: a nationwide field study including 19 sterilization departments” may be accepted after major revision. The comments are as follows:

1. The language of the manuscript should be improved throughout.

2. The authors need to strengthen the scientific interpretations of the facts thoroughly.

3. Disinfection processes for reuse have not been described for all types of masks.

4. Studies not showing how to remove retention of the virus in the mask.

6. PLOS authors have the option to publish the peer review history of their article (what does this mean?). If published, this will include your full peer review and any attached files.

Reviewer #1: No

Reviewer #3: No

---

## [Author Response · Author response to Decision Letter 0]

7 Aug 2021

We would like to thank all of the reviewers for their comments. This round of revision resulted in valuable feedback from the reviewers. We have processed the feedback accordingly and carefully addressed each comment.

We fully agree with the comments raised and improved the manuscript to be more technically sound, with more thorough analysis and improved English grammar. Please find our responses in the document 'Response to Reviewers'.

---

## [Decision Letter · Decision Letter 1]

2 Sep 2021

Can sterilization of disposable face masks be an alternative for imported face masks? A nationwide field study including 19 sterilization departments and 471 imported brand types during COVID-19 shortages.

PONE-D-21-13083R1

Dear Dr. van Straten,

We’re pleased to inform you that your manuscript has been judged scientifically suitable for publication and will be formally accepted for publication once it meets all outstanding technical requirements.

Kind regards,

Mahendra Singh Dhaka, Ph.D.

Academic Editor

PLOS ONE
---

## [Editor Report · Acceptance letter]

6 Sep 2021

PONE-D-21-13083R1 

Can sterilization of disposable face masks be an alternative for imported face masks? A nationwide field study including 19 sterilization departments and 471 imported brand types during COVID-19 shortages. 

Dear Dr. van Straten:

I'm pleased to inform you that your manuscript has been deemed suitable for publication in PLOS ONE. Congratulations! Your manuscript is now with our production department. 

Kind regards, 

on behalf of

Dr. Mahendra Singh Dhaka 

Academic Editor

PLOS ONE